# Peer review of "High Prevalence of Strongyloidiasis in Spain: A Hospital-Based Study"

_pathogens, 2020, doi:10.3390/pathogens9020107_

Round 1

Reviewer 1 Report

Dear author,

This is a well-written and important study.

Access to ivermectin for strongyloidiasis in general practice settings  is imperative. Treating chronic strongyloidiasis is a primary health care role.

As a Reviewer, I would have the following remarks regarding the manuscript:

The inclusion criteria should be improved(details about the immunosuppression risk). Please use "Italic" style when you write the name of the parasites.

The manuscript has also some grammar mistakes (especially in the introduction part), and some sentences that need to be reformulate. That is why, the English written style should be revised by a English native speaker.

Best regards!

Author Response

We really appreciate all the comments'  reviewer. We have addressed them one by one.

The inclusion criteria should be improved (details about the immunosuppression risk). Please use "Italic" style when you write the name of the parasites.

We thanks the reviewer the comment. Unfortunately, we had only information about the immunosuppression risk in the group of people with a positive serological test. We have included additional information in the methods and in the results concerning the immunosuppression risk.

Concerning the italics, apologies. Indeed it was a mistake generated when we changed to the "Pathogens format". Now, it has been addressed.

RESULTS: Page 4 line 137 (track changes version)

“There were 33/176 (18.75%) individuals with Strongyloides that were considered immunosuppressed. There were 3 transplant patients, 3 patients with neoplasia, 10 patients with an autoimmune disease, 2 patients under treatment with steroids for other reasons, and 14 patients with HIV and with <500 CD4 (one of whom had <200 CD4). There were an additional 17 patients with HIV and >500 CD4 that were not considered immunosuppressed individuals”.

METHODS. Page 7, line 266 (track changes version)

In particular, infectious diseases or tropical disease units attending to potentially severe immunosuppressed patients (Hematology, Oncology, autoimmune diseases/Rheumatology, and HIV and transplant units) were willing to participate in and implement the systematic screening. In these units, the screening was conducted irrespective of whether the individual was immunosuppressed or not. However, it should be added that the reasons for hospitalization in those units were always conditions related to such units. Not all hospital units or departments participated in the study.

METHODS. Page 8. Line 284

“All positive patients were invited to participate, signed the consent form, and were provided treatment with ivermectin 200 mcg/kg over two consecutive days.  Information on aspects related to immunosuppression and other clinical and laboratory aspects were collected in a questionnaire. Participants were clinically and microbiologically followed-up with a serological test and with three stool tests if the initial stool test was positive after six months of therapy as part of the routine clinical practice.”.

2. The manuscript has also some grammar mistakes (especially in the introduction part), and some sentences that need to be reformulate. That is why the English written style should be revised by a English native speaker.

We thanks the reviewer this comment. Accordingly, the manuscript has been reviewed by an English native person (Martyn Rittman, Ph.D. English Editing Manager).

Reviewer 2 Report

Good overall premise.  A general concern regarding the assumption that patients present in the "immunosuppressed" units (autoimmune, transplant heme/onc, HIV) are actually immunosuppressed.  Feels like a big jump to assume that patients being treated on those units are actually that population.  The word "potentially" is used in the abstract (line 45) that adds to the suspicion.  Perhaps that could be addressed more in the introduction or methods section.  For example, elaborate on how patients must have those conditions to be located in those units if that is the case for the 6 hospitals or how frequently it occurs that patients are on those units without the specified condition.  In the US, we admit patients to unrelated units if the hospital census is high and there is not a bed located on a unit that the patient would have been sent to if there was a bed available (e.g. a medical pneumonia patient could be admitted to the orthopedic unit and does not have any current orthopedic issue).

Page 1, Abstract, there is a suggestion that prevalence data are scarce for migrant populations.  However, the Lancet publication meta-analysis of migrants  originating  from endemic countries who are living in host countries with  low  prevalencemigrants from endemic countries entering countries with low prevalence from 2019 identified 84 studies with stool and serological surveillance for a total of 72,000+ people.  This doesn't seem to be very scarce.

Page 2, line 54, add "Human" to migration flows to clarify what population.  Also, add "Latin" to the American description

Page 2, line 64 add the concept of "autoinfection" to the lifetime infection of the host description

Page 2, lines 74-76 sentence starting with "In this regard..." does not seem to add anything to the previous line, consider deleting the sentence.

Page 2, line 77, consider deleting "where re-infection is unlikely", that would be implied for a non-endemic area

Results - I did not see whether any patients went on to develop hyperinfection or disseminated infections in the results section.  Should this be addressed whether or not it was captured or occurred and if not, why?  

Page 5 Figure 1, add "Identified" seroprevalence of strongyloidiasis in migrants.... to emphasize that this is the current study results and not the general understood seroprevalence

Page 6, line 190 double negative = "not negligible" consider rewording for better clarity

Page 6, lines 192-194 sentence "Although the limited..." sentence is confusing and request clarification

Page 6, lines 195-200 Paragraph is confusing and request rewrite

Page 6, line 203, consider adding a theory as to why the seroprevalence was higher for women in your study

Page 6, line 205-206 - related to the general concern that location in a unit is an assured diagnosis, add whether or not that is an issue into the description.

Page 7, line 212 another double negative to consider rewording "not clear if serology should not be..."

Page 7, line 229, you reference a lack of systematic screening that overestimated the prevalence.  However, isn't the premise of your study that you did utilize systematic screening in the specific units of focus?  Clarify this statement

Page 7, line 244, I am still confused on how specifically patients were selected for screening.  Was it up to the attending physician to note that the patient was a migrant from an endemic area and a manual process to remember to order the serological test?  Or was there an automated process?

Page 7, line 254.  Elaborate on how patients were microbiologically followed six months after therapy.  Was the ELISA test resent?

Page 8, line 258. Consider adding the sensitivity/specificity of the specific test you utilized for all patients

Page 8, line 290. Consider rewording your conclusion regarding the immunosuppressed population as your findings did not identify an increased risk of being positive.  Or consider elaborating to say that even though you didn't find a higher rate in this population, that the test is not as sensitive and considering the clinical impact of hyperinfection or disseminated infection in this population, they should always be considered for screening regardless.

Page 8, table 1, add "Latin" in front of America.  And clarify that your prevalence is in % or whatever the unit is you are listing prevalence as

Also, review countries listed in table 1 for English translation (e.g. Rumania = Romania; Ukraina = Ukraine, not sure if there are others)

Author Response

We really appreciate all comments' reviewer. We have addressed one by one. Please find below them.

1. Good overall premise.  A general concern regarding the assumption that patients present in the "immunosuppressed" units (autoimmune, transplant hame/onc, HIV) are actually immunosuppressed.  Feels like a big jump to assume that patients being treated on those units are actually that population.  The word "potentially" is used in the abstract (line 45) that adds to the suspicion.  Perhaps that could be addressed more in the introduction or methods section.  For example, elaborate on how patients must have those conditions to be located in those units if that is the case for the 6 hospitals or how frequently it occurs that patients are on those units without the specified condition.  In the US, we admit patients to unrelated units if the hospital census is high and there is not a bed located on a unit that the patient would have been sent to if there was a bed available (e.g. a medical pneumonia patient could be admitted to the orthopedic unit and does not have any current orthopedic issue).

 We thank the reviewer this comment. We have included more information in the method section. In addition, we would like to add that in all hospitals included in the study the “Unit” to which a patient is admitted is not physically located in a same place, which mains (and following your example) that a medical pneumonia patient can be hospitalized in the orthopedic floor, although the admission is in the Infectious disease unit. We call this: “Ectopic patient” but essentially the unit will be the one as the medical doctor belongs to. An therefore, screening has been implemented by units and not “floors” at the hospital. I hope this clarifies you your point. We have also introduced some aspects in the text.

Methods section. Page 7 Line 263 (track changes version)

“Although systematic screening has not been widely implemented at a national level, it was established that individuals either hospitalized or being attended to in any specialized out-patient or in-patient units of the hospital should be screened for strongyloidiasis. In particular, infectious diseases or tropical disease units attending to potentially severe immunosuppressed patients (Hematology, Oncology, autoimmune diseases/Rheumatology, and HIV and transplant units) were willing to participate in and implement the systematic screening. In these units, the screening was conducted irrespective of whether the individual was immunosuppressed or not. However, it should be added that the reasons for hospitalization in those units were always conditions related to such units. Not all hospital units or departments participated in the study”.

2. Page 1, Abstract, there is a suggestion that prevalence data are scarce for migrant populations.  However, the Lancet publication meta-analysis of migrants originating  from endemic countries who are living in host countries with  low  prevalence migrants from endemic countries entering countries with low prevalence from 2019 identified 84 studies with stool and serological surveillance for a total of 72,000+ people.  This doesn't seem to be very scarce.

We agree the author but what we wanted to highlight is the scarcity of seroprevalence data (total sample size around 10000 people), particular in Asian countries. We have modified the text in the following way:

Abstract. Page 1. Line 36 (track changes version)

 “Strongyloidiasis is a prevailing helminth infection ubiquitous in tropical and subtropical areas. However, seroprevalence data are scarce in migrant populations, particularly for those coming for Asia”.

Introduction. Page 2. Line 91 (track changes version ).

“Evidence of the seroprevalence of strongyloidiasis in migrant populations  is scarce, particularly if it is assessed in Asian countries.  although itAvailable data suggest that it is known to vary substantially, depending on the country of origin, being particularly high in people coming from countries such as Cambodia (36%) or Latin American countries (26%) [21]”.

3. Page 2, line 54, add "Human" to migration flows to clarify what population.  Also, add "Latin" to the American description

Agreed and accordingly modified.

4. Page 2, line 64 add the concept of "autoinfection" to the lifetime infection of the host description

We have added the following information in the Introduction. Page 2, line 64 (track changes version)

“Firstly, the infection can persist for the whole lifetime due to the possibility of causing autoinfection in the human host [8]. Through this phenomenon, the filariform larvae penetrate intestinal mucosa in the large intestine or perianal skin and migrate to complete another lifecycle.”

5. Page 2, lines 74-76 sentence starting with "In this regard..." does not seem to add anything to the previous line, consider deleting the sentence.

Agreed and accordingly modified

6. Page 2, line 77, consider deleting "where re-infection is unlikely", that would be implied for a non-endemic area

Agreed and accordingly modified

7. Results - I did not see whether any patients went on to develop hyperinfection or disseminated infections in the results section.  Should this be addressed whether or not it was captured or occurred and if not, why?  

We thank the reviewer for this comment. We have added in the results section, the following information (Page 4, line 136; track changes version)

“There were 33/176 (18.75%) individuals with Strongyloides that were considered immunosuppressed. There were 3 transplant patients, 3 patients with neoplasia, 10 patients with an autoimmune disease, 2 patients under treatment with steroids for other reasons, and 14 patients with HIV and with <500 CD4 (one of whom had <200 CD4). There were an additional 17 patients with HIV and >500 CD4 that were not considered immunosuppressed individuals. No patients developed hyperinfection or disseminated disease. ”.

8. Page 5 Figure 1, add "Identified" seroprevalence of strongyloidiasis in migrants.... to emphasize that this is the current study results and not the general understood seroprevalence

Agreed and accordingly modified

9. Page 6, line 190 double negative = "" consider rewording for better clarity

Agreed and accordingly modified:

10. Section Discussion. Page 6. Line 206 (Track changes version)

“… our data show a not negligible considerable seroprevalence rate, being with seroprevalence higher than 5% in countries such as Morocco or the Philippines”.

11. Page 6, lines 192-194 sentence "Although the limited..." sentence is confusing and request clarification

12. Page 6, lines 195-200 Paragraph is confusing and request rewrite

We agree with the reviewer’s comment. We have rewritten the whole paragraph (Section Discussion, page 6, line 205; Track changes version) in the following way:

“In South-East Asia and North Africa, our data show a considerable seroprevalence rate, being higher than 5% in countries such as Morocco or the Philippines. However, the small number of individuals screened coming from these countries is very low, which may prevent proper conclusions from being drawn regarding the seroprevalence in these geographic areas. Published data on the Strongyloides prevalence in Asian countries are very scarce and inexistent in other countries (Morocco and the Philippines). Although we have not found studies specifically providing prevalence estimates in Asian migrants, some surveillance data about health conditions in migrants have reported a 5%–7% rate of strongyloidiasis among people coming from South-East Asia and South Asian countries[31]. Further studies conducted in non-endemic areas should better evaluate the S.stercoralis prevalence in people coming from Asian countries”.

13. Page 6, line 203, consider adding a theory as to why the seroprevalence was higher for women in your study

We have added the following information in the Discussion section (Page 7, line 223; track changes version):

“One possible explanation for this is the large contribution of Bolivian women to the population tested in our dataset, which have a high seroprevalence”.

14. Page 6, line 205-206 - related to the general concern that location in a unit is an assured diagnosis, add whether or not that is an issue into the description.

We already clarified in a previous comment that location in a unit is an assured risk of having a related condition although not necessarily an immunosuppression. We have clarified it in the text in the following way (Discussion section, page 7, line 228; track changes version):

“In our study, the prevalence in HIV-infected individuals and patient from transplant units, but not necessarily immunosuppressed individuals, was much lower compared with other general out /in-patients units services of the hospitals”.

15. Page 7, line 212 another double negative to consider rewording "not clear if serology should not be..."

We have rephrased the paragraph in the following way (Discussion section, page 7, line 236; track changes version):

“In addition, the accuracy of the serological test in the diagnosis of strongyloidiasis in immunosuppressed patients, and which is the best strategy for screening in such cases, is still unclear”.

16. Page 7, line 229, you reference a lack of systematic screening that overestimated the prevalence.  However, isn't the premise of your study that you did utilize systematic screening in the specific units of focus?  Clarify this statement.

We thank the reviewer this comment. We would like to clarify that although our objective was to conduct a systematic screening in all these units, we cannot guarantee that all medical doctors from all units have performed the screening to all at-risk-patients risk (selection bias). We have introduced the following information in the manuscript (Discussion section, page 7, line 255; track changes version )

“However, a selection bias may have been introduced if the screening was not systematic in all at-risk-individuals, which could have overestimated the prevalence in some units”.

17. Page 7, line 244, I am still confused on how specifically patients were selected for screening.  Was it up to the attending physician to note that the patient was a migrant from an endemic area and a manual process to remember to order the serological test?  Or was there an automated process?

We have clarified this question in the following way (Discussion section, page 8, line 274; track changes version).

“For that purpose, training sessions were organized in those units attending to potentially immunosuppressed patients to enhance the importance of Strongyloides screening and the physicians in charge of the patient ordered the serological test”.  

18. Page 7, line 254.  Elaborate on how patients were microbiologically followed six months after therapy.  Was the ELISA test resent?

We have clarified it adding the following information (page 8, line 277, track changes version)

“Participants were clinically and microbiologically followed-up with a serological test and with three stool tests if the initial stool test was positive after six months of therapy as part of the routine clinical practice”.

19. Page 8, line 258. Consider adding the sensitivity/specificity of the specific test you utilized for all patients

We have added the following information in the manuscript (Discussion section, page 8, line 298; track changes version)

“The sensitivity and specificity for this test using this cut-off have been reported to be 91.2% and 99.1% respectively [33]”.

Bisoffi Z, Buonfrate D, Sequi M, Mejia R, Cimio RO, Krolewiecki AJ, et al. Diagnostic accuracy of five serologic tests for Strongyloides stercoralis infection. PLoS Negl Trop Dis. 2014 Jan 9;8(1):e2640. doi: 10.1371/journal.pntd.0002640. eCollection 2014

20. Page 8, line 290. Consider rewording your conclusion regarding the immunosuppressed population as your findings did not identify an increased risk of being positive.  Or consider elaborating to say that even though you didn't find a higher rate in this population, that the test is not as sensitive and considering the clinical impact of hyperinfection or disseminated infection in this population, they should always be considered for screening regardless.

We thanks the reviewer this comments. We have rephrased the conclusions in the following way (Conclusions section, page 9, line 325; track changes version):

“In this paper, we reported a hospital-based systematic screening of Strongyloides with a seroprevalence of almost 10% in the migrant population from endemic areas, and thus presented evidence for the need to implement strongyloidiasis screening strategies in hospitalized patients. Even though such a high rate was not found in the potentially immunosuppressed group of people, screening should be particularly considered in these cases due to the clinical impact of hyperinfection or disseminated infection in this population, and also because of a lack of accurate data on serological testing in immunosuppressed people.”.

21. Page 8, table 1, add "Latin" in front of America.  And clarify that your prevalence is in % or whatever the unit is you are listing prevalence as

Agreed and accordingly modified

22. Also, review countries listed in table 1 for English translation (e.g. Rumania = Romania; Ukraina = Ukraine, not sure if there are others)

Thanks for the observation. We have checked the whole list of countries.

Reviewer 3 Report

Well designed, analysed, and presented. 

Author Response

We really thank you the review her/his positive impression of the manuscript.

We attach the edited version including the modifications required by other reviewers.
